# Stochastic Adaptive Activation Function

**Kyungsu Lee**
DGIST*
42988 Daegu, South Korea
ks_lee@dgist.ac.kr

**Jaeseung Yang**
DGIST
42988 Daegu, South Korea
yjs6813@dgist.ac.kr

**Haeyun Lee**
DGIST, SAMSUNG SDI
17084 Yong-In, South Korea
haeyun.lee@samsung.com

**Jae Youn Hwang**[†]
DGIST
42988 Daegu, South Korea
jyhwang@dgist.ac.kr

## Abstract

The simulation of human neurons and neurotransmission mechanisms has been realized in deep neural networks based on the theoretical implementations of activation functions. However, recent studies have reported that the threshold potential of neurons exhibits different values according to the locations and types of individual neurons, and that the activation functions have limitations in terms of representing this variability. Therefore, this study proposes a simple yet effective activation function that facilitates different thresholds and adaptive activations according to the positions of units and the contexts of inputs. Furthermore, the proposed activation function mathematically exhibits a more generalized form of Swish activation function, and thus we denoted it as Adaptive SwisH (ASH). ASH highlights informative features that exhibit large values in the top percentiles in an input, whereas it rectifies low values. Most importantly, ASH exhibits trainable, adaptive, and context-aware properties compared to other activation functions. Furthermore, ASH represents general formula of the previously studied activation function and provides a reasonable mathematical background for the superior performance. To validate the effectiveness and robustness of ASH, we implemented ASH into many deep learning models for various tasks, including classification, detection, segmentation, and image generation. Experimental analysis demonstrates that our activation function can provide the benefits of more accurate prediction and earlier convergence in many deep learning applications. [1]

## 1 Introduction

Searching for the optimal activation functions has been a challenge in the field of artificial intelligence (Maas et al., 2013; Ramachandran et al., 2017; Clevert et al., 2015). Early activation functions have been studied to compensate for the non-linearity of the artificial neural networks or ameliorate the gradient vanishing problem (Hertz et al., 1997; Hochreiter, 1998). Recently, novel activation functions have been suggested with the zero-centered or parametric properties that improve the training efficiency of deep neural networks (DNNs) (Maas et al., 2013; Clevert et al., 2015). Currently, the activation functions focusing on the stability of DNNs or probabilistic distribution of inputs have been proposed (Hendrycks, Gimpel, 2016; Misra, 2019). Advances in the activation functions have

---

*Daegu Gyeongbuk Institute of Science and Technology

†corresponding author

[1]Our code is available at https://github.com/kyungsu-lee-ksl/ASH

allowed DNNs to perform various tasks such as detecting or segmenting target objects in sophisticated images or even generating new images beyond the simple classifiers (Simonyan, Zisserman, 2014; Zhao et al., 2019; Badrinarayanan et al., 2017; Goodfellow et al., 2014).

The activation functions has evolved to behave more like a human neuron (Sharma et al., 2017; Lee et al., 2017). However, Izhikevich (2003); Evans et al. (2018) reported that the neurotransmission mechanism, including the membrane, action, and threshold potentials of human neurons, is subject to the location or the connection type of the neurons. Additionally, humans perceive objects with surrounding contexts using the $N : N$ mapping of visions to human neurons rather than the 1:1 mapping of a pixel to an input node in a neural network (Liu et al., 2018). The connections between neurons can be realized through the linear combinations of layers in DNNs. However, DNNs have limitations in terms of realizing contextual perception. This implies that the further improvement in deep neural networks and convolutional neural networks (CNNs) can be realized despite the impressive performance on image analysis (Jinsakul et al., 2019; Misra, 2019). Therefore, the development of DNNs is leaned to mimic human perception by realizing the mechanism of human neurons (Aggarwal, others, 2018; Lindsay, 2021).

tCurrently, the primary issue is that many activation functions exhibit passivity, in this paper, indicating that they determine outputs only concerning the value of one element rather than entire contexts. For instance, the Rectified Linear Unit (ReLU), defined as $f(x) = \max(x, 0)$, determines the output values related to $x$ (Fukushima, Miyake, 1982), whereas the *softmax* function generates output values as the ratio of the input value to the totals (Goodfellow et al., 2017). Particularly, ReLU exhibits passivity, whereas *softmax* does not. Suppose an image can be classified by considering 80% of the total portion. Current activation functions are limited in terms of classifying such an image since only elements (pixels) of the image are used to rectify the image rather than a ratio. Another limitation is that the activation functions are invariant. Although the parametric activation functions update their parameters during training, the resulting parameters are invariant during the inference phase (Xu et al., 2015; Bingham, Miikkulainen, 2022). Therefore, the limited rectification can be realized by the invariant parameters or thresholds regardless of new inputs from different domains (e.g., test set).

**Contribution**  To realize the mechanism of human neurons that rectify inputs considering their contexts, we propose a novel ASH activation function. The main contributions of this study are to suggest a simple yet effective activation function, ASH, and to implement ASH in a mathematically effective form. Going beyond the passive activation functions, ASH activation function is designed as (1) an active activation function that provides outputs regarding the context of inputs and (2) a conditional activation function that employs an adaptive threshold. Unlike ReLU or Leaky ReLU, the threshold value of ASH is adaptively changed by considering the contextual information.

$$f(x) = \begin{cases} x & \text{if } x \geq \theta, \\ 0 & \text{otherwise} \end{cases} \tag{1}$$

In particular, the threshold value ($\theta$) is adaptively changed according to the input distribution without heavy calculation, and thus ASH provides outputs considering the contexts of inputs adaptively. By applying ASH, we obtained the following theoretical and experimental results:

- We conducted mathematical modeling on ASH in an effective form to ensure trainable and parametric properties, and thus ASH exhibits parametric and adaptive properties. The baseline threshold of ASH is initially trained during the training phase, and the threshold value is adaptively fine-tuned according to the contexts of inputs without heavy calculations.

- We theoretically verified that ASH adaptively changes its threshold alongside the stochastic distribution of inputs. This implies that ASH provides outputs regarding the entire contexts of inputs, thus leading to enhanced feature extraction.

- We theoretically verified that ASH exhibits a generalized formula of Swish activation function and provided the mathematical explanations for the superior performance of Swish, which was empirically searched in the previous work.

- We experimentally showed that ASH improves the performance of deep learning models on various tasks and shortens the convergence epoch.

**Related works**  Activation functions affect the performance of the training process to determine a functional subspace of a DNN (Hayou et al., 2019). In particular, the non-linearity using activation

functions have been introduced to prevent the issue of the linear transformation causing simple feature extractions in the DNNs (Misra, 2019; Jarrett et al., 2009). DNNs with non-linearity have been employed to perform complex tasks (Leshno et al., 1993). In the early era, Rectified Linear Unit (ReLU) replaced the classical activation functions such as sigmoid and tanh (Nair, Hinton, 2010) due to its simple and computational efficiency compared to other activation functions. During decades, many activation functions have been proposed , including Leaky ReLU (Maas et al., 2013), Exponential Linear Unit (ELU) (Clevert et al., 2015), Gaussian Error Linear Unit (GELU) (Hendrycks, Gimpel, 2016), Scaled Exponential Linear Unit (SELU) (Klambauer et al., 2017), and Swish (Ramachandran et al., 2017) to improve the performance and stability of learning parameters in DNNs. Those activation functions have solved the dying ReLU problem, which exhibits a zero value in the negative region, and improved DNNs more smoothly for stable optimization. In particular, ELU and SELU have realized internal normalization in the layer using the zero-mean property. GELU exploited a Gaussian error and could implement an adaptive dropout to apply a higher probabilistic intuition.

**Problem statement** For adaptive thresholding, ASH exploits a stochastic selection methodology such as a quick selection (Hoare, 1961). In particular, for enhanced feature extraction, the informative elements, which exhibit large values, should be identified as an attention mechanism. In contrast, some elements, which exhibit low relevance, should be required to be rectified. To this end, ASH is designed to identify informative elements but rectify others as 0.

Let $X \in R^{H \times W \times C}$ be a tensor (i.e., feature-map) disregarding the batch, but with a height ($H$), width ($W$), and channel ($C$), and let $\bar{X} \ni X$ be a set of feature-maps. Let $\mathcal{A}$ be an activation function such that $\mathcal{A} : \bar{X} \to \bar{X}$. We can then extract the $i^{th}$ element from $X$, and denote it as $x^{(i)}$. Here, the goal of this study is to design the activation function represented as follows:

$$\mathcal{A}(x^{(i)}) = \begin{cases} x^{(i)} & \text{if } x^{(i)} \text{ is ranked in the top-k percentile of X,} \\ 0 & \text{otherwise} \end{cases} \qquad (2)$$

Note that, the novel activation function $\mathcal{A}$ is represented similar to ReLU, whereas its threshold is not invariant in contrast to ReLU, but it is subjected to the distribution of the input $X$. To simplify, let $C(x^{(i)}, k; X)$ be a condition whether $x^{(i)}$ is ranked in the top $k$ percentile of $X$, and the negation of $C$ is denoted as $\neg C$. In particular, the elements that satisfy $C(x^{(i)}, k; X)$ are from the first largest element to the $(0.01kN)$-th largest element in $X$, where $N$ indicates the number of elements in $X$. Here, the elements can be extracted using a simple algorithm as a quick selection. Suppose a set $\hat{X}$ that includes all elements in $X$ such that $\hat{X} = \{x^{(1)}, x^{(2)}, ..., x^{(N)}\}$. We can then construct subsets of $\hat{X}$ as $\hat{X}_C = \{x^{(i)} \in X | C(x^{(i)}, k; X)\}$ and $(\hat{X}_C)^c = \{x^{(i)} \in X | \neg C(x^{(i)}, k; X)\}$. Equation (2) can be then simplified as follows:

$$\mathcal{A}(x^{(i)}) = \begin{cases} x^{(i)} & \text{if } x^{(i)} \in \hat{X}_C, \\ 0 & \text{otherwise} \end{cases} \qquad (3)$$

In summary, the activation function is designed to sample the top-$k$ percentile from the input, where criteria are the values of elements. Sampling examples are presented in Fig. 1, compared to ReLU.

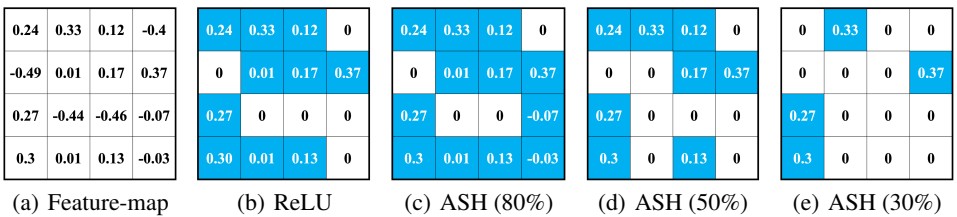

| (a) Feature-map | (b) ReLU | (c) ASH (80%) | (d) ASH (50%) | (e) ASH (30%) |

Figure 1: Input feature-map and outputs by activation functions. ASH ($k\%$) indicates that ASH activation function samples top-$k\%$ elements from input feature-map. The sampled elements by the activation functions are colored as blue.

## 2 Method

This study aims to design an activation function for a stochastic sampling of the top-$k$ percentile elements from inputs. However, sorting or sampling methods such as a quick selection requires high computational costs. In contrast, sampling the top-$k$ percentile can be realized using a Z-score-based method in a simple calculation despite the prerequisites of a normal distribution (also known as Gaussian distribution). As discussed below, the outputs of neurons are normally distributed. Therefore, we employed the stochastic sampling to design ASH activation function.

In this section, we (1) demonstrate that the outputs of neurons are normally distributed, (2) construct a model for stochastic sampling using a Z-score, (3) formulate ASH activation function, (4) verify that ASH is parametric and trainable, and (5) search for general applications of ASH activation function.

### 2.1 Background

**Gaussian distribution**   Many deep learning models employ normalization methods to improve their stability in training (Ioffe, Szegedy, 2015; Ulyanov et al., 2016; Wu, He, 2018). In a previous study, Ioffe, Szegedy (2015) reported that the output of the convolutional layer, $x = Wu + b$, is more likely to have a symmetric, non-sparse distribution, that is "more Gaussian". Since most deep learning models are based on convolutional operations, the outputs of neurons are supposed to be normally distributed (Gaussian distribution). Therefore, it is concluded that the inputs of the activation functions are normally distributed when activation functions follow convolutional layers, such that $x \sim N(\mu_x, \sigma_x^2)$, where $x$ is an input feature-map of an activation function, $\mu_x$ and $\sigma_x$ are mean and standard deviation of $x$, respectively. Therefore, we can obtain the following proposition.

**Proposition 1.** The outputs of neurons in convolutional neural networks are normally distributed.

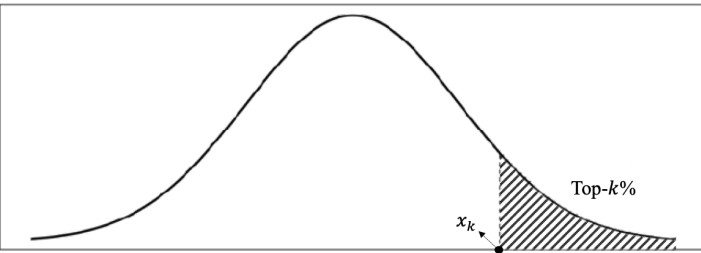

Figure 2: Top-$k$% sampling from normal distribution

**Sampling from a normal distribution**   This study aimed to sample the top-$k$% elements that exhibit large values in an input feature-map. Since the area under the normal distribution indicates the percentile, sampling the top-$k$% from a normal distribution is theoretically identical to the statement calculating the area under the curve presented in Fig. 2. Let $F$ be a tensor, a multi-dimensional array or a matrix. $F$ should be then normally distributed, and thus we can sample the elements ($f^{(i)}$) in the top-$k$% from $F$ using the following equation:

$$P(f^{(i)} \geq z_k') = k, \ \ k \in [0,1] \ \ z_k' \in [-\infty, \infty] \tag{4}$$

However, heavy computational costs are incurred to obtain a trivial solution from the probability density function of the normal distribution, defined as $\frac{1}{\sigma_F \sqrt{2\pi}} e^{-(x-\mu_F)^2/2\sigma_F^2}$. Therefore, we leaned to *probability theory* to simplify the computation rather than *calculus*. To this end, we employed the standard normal distribution (Z-score normalization) for Equation (4), and we obtained the following equation:

$$P(Z^{(i)} \geq z_k) = k, \ \ Z^{(i)} = \frac{f^{(i)} - \mu_F}{\sigma_F} \ \ s.t. \ \ Z^{(i)} \sim N(0,1) \tag{5}$$

where $z_k = (z_k' - \mu_F)/\sigma_F$, which is the Z-normalized value from $z_k'$, and thus $z_k$ is subjected to $k$, indicating percentile to sample, in terms of Z-table (Larsen, Marx, 2005). Then, we go Z-table and

easily find the proper value for $z_k$, intuitively. Therefore, the condition, $Z^{(i)} = (f^{(i)} - \mu_F)/\sigma_F \geq z_k \Leftrightarrow f(i) \geq \mu_F + z_k\sigma_F$, is mathematically identical to sample the top-$k\%$ elements from $F$. To summarize, we obtained the following proposition.

**Proposition 2.** Element $x^{(i)} \in X$, which is normally distributed, is in the top-$k\%$ of $X$ if $x^{(i)} \geq \mu_X + z_k\sigma_X$, where $z_k$ is a z-value subjected to $k$ in Z-table (Larsen, Marx, 2005).

**Differentiation** The mechanisms of convolutional neural networks (CNNs) have been studied in many previous works (Rumelhart et al., 1986; Bottou, 2010; Zhang, 2016; Hu et al., 2018). Training CNN models is subjected to the backpropagation derived from the partial derivatives of loss functions by the individual convolutional parameters. Let $L$ be a loss function for the deep learning model $M$ and let $W$ be one of the variables in $M$. Then, the derivative of $L$ in terms of $W$ is represented as $\frac{\partial L}{\partial W}$, and $W$ is updated as $W \leftarrow W - \eta\frac{\partial L}{\partial W}$ with a learning rate $\eta$. On the other hand, suppose variable $\theta$ be the threshold, the function $f(x)$ is $\alpha x$ if $x \geq \theta$, otherwise 0. Thus, the partial derivative of $f$ is represented as follows:

$$\frac{\partial f}{\partial x} = \begin{cases} \alpha, \text{ if } x \geq \theta \\ 0, \text{ otherwise} \end{cases} \quad , \quad \frac{\partial f}{\partial \alpha} = \begin{cases} x, \text{ if } x \geq \theta \\ 0, \text{ otherwise} \end{cases} \quad , \quad \frac{\partial f}{\partial \theta} = 0 \tag{6}$$

Here, $\alpha$ is arithmetically combined with $f(x)$, whereas $\theta$ does not. Therefore, $\alpha$ is trainable, but $\theta$ is not trainable in this context. In basic *calculus*, it is trivial that if the loss function is not dependent on the variable, the partial derivative is zero, and thus the variable cannot be trained or optimized; In other words, it is invariant. Therefore, we obtain the following lemma:

**Lemma 1.** The derivative of a variable in the conditional statement is zero, and thus that the variable cannot be optimized.

## 2.2 ASH Activation Function

Let $X$ be an input of ASH activation function ($\mathcal{E}$) and be a tensor of which elements are normally distributed. Furthermore, let $x^{(i)} \in X$ be the $i$-th element in $X$. Then, by **Proposition 2**, ASH activation function, which samples the top-$k\%$ elements from the input, is represented as follows:

$$\mathcal{E}(x^{(i)}) = \begin{cases} x^{(i)} & \text{if } x^{(i)} \geq \mu_X + z_k\sigma_X, \\ 0 & \text{otherwise} \end{cases} \tag{7}$$

where $\mu_X$ and $\sigma_X$ are the mean and the standard deviation of all elements in $X$, respectively, and $z_k$ is the Z-score concerning percentile ($k$) to sample (i.e., $z = 1.96$ if $k = 2.5\%$, see Z-table). Equation (7) exhibits that ASH activation function is represented in a simple yet effective form with low computational costs.

Intuitively, we assumed that the activation level (percentile) is supposed to be different by each neuron and the tasks of the deep learning model, similar to human neurons. However, in Equation (7), the condition ($x^{(i)} \geq \mu_X + z_k\sigma_X$) is invariant by **Lemma 1**. Note that, $\mu_X + z_k\sigma_X$ is variable and changeable with respect to $X$, but the sampled portion ($k\%$ related to $z_k$) from $X$ is invariant. Therefore, ASH in Equation (7) is not parametric and trainable. To make ASH be trainable and parametric, let Equation (7) be substituted using a proxy function as $\mathcal{E}(x^{(i)}) = x^{(i)}f(x^{(i)})$, such that the proxy function $f(x^{(i)})$ is represented as follows:

$$f(x^{(i)}) = \begin{cases} 1 & \text{if } x^{(i)} - \mu_X - z_k\sigma_X \geq 0, \\ 0 & \text{otherwise} \end{cases} \tag{8}$$

For simplicity, suppose that a Heaviside step function (Weisstein, 2002) is defined as $H(x) = \frac{d}{dx}\max(0, x)$, and thus $f(x^{(i)}) = H(x^{(i)} - \mu_X - z_k\sigma_X)$. Then, we obtain the arithmetical form to formulate ASH activation function as follows:

$$\mathcal{E}(x^{(i)}) = x^{(i)}H(x^{(i)} - \mu_X - z_k\sigma_X) \tag{9}$$

Even with the arithmetic formula, ASH activation function in Equation (9) is still independent of $z_k$, and thus the $z_k$ is still not trainable. However, it is well known that the Heaviside step function is

analytically approximated as $2H(x) = 1 + 1\tanh(\alpha x)$ with a large value of $\alpha$ (Iliev et al., 2017), and thus ASH activation function is approximated using the smooth Heaviside step function as follows:

$$\begin{aligned}
\text{Æ}(x^{(i)}) &= x^{(i)}H(x^{(i)} - \mu_X - z_k\sigma_X) \\
&= \frac{1}{2}x^{(i)} + \frac{1}{2}x^{(i)}\tanh(\alpha(x^{(i)} - \mu_X - z_k\sigma_X)) \\
&= \frac{x^{(i)}}{1 + e^{-2\alpha(x^{(i)} - \mu_X - z_k\sigma_X)}}
\end{aligned} \tag{10}$$

Since $z_k$ is arithmetically placed, $z_k$ representing a sampling percentile is trainable, and thus ASH activation function is also trainable and parametric. By optimizing $z_k$, ASH activation functions exhibit different thresholds. Therefore, it is concluded that ASH exhibits different activation levels based on the stochastic sampling of inputs and different thresholds, similar to human neurons, synapses, and their potentials. As human neurons, the mechanism of ASH can be summarized as follows:

(1) In the training phase, each ASH activation function optimizes its $z_k$ and fine-tunes the threshold for the sampling percentile of inputs. Thus, it implies that ASH activation function realizes the arbitrary threshold potentials as human neurons (Clevert et al., 2015; Evans et al., 2018). Some examples of $z_k$ related to Equation (7) are:

**Example 1.** A small value of $z_k$, even a small negative value, implies the dense activation, and the dying ReLU problem can be solved.

**Example 2.** A large value of $z_k$ implies the sparse activation, and the sparsity can be leveraged.

(2) In the training or inference phase, ASH activation function rectifies the inputs using the learned threshold value and contexts of inputs. In particular, to sample the top-$k$ percentile, ASH employs the mean and the standard deviation of inputs, representing the contexts of the entire inputs. Therefore, it implies that ASH realizes the adaptive activation considering the contexts of inputs. Some examples of the adaptive activation related to Equation (7) are:

**Example 3.** A small threshold value ($\theta_s$) is employed to calculate the input $X$ that exhibits large mean and standard deviation values ($X > \theta_s$).

**Example 4.** A large threshold value ($\theta_l$) is employed to calculate the input $X'$ that exhibits large mean and standard deviation values ($X' > \theta_l \gg \theta_s$).

### 2.3 Generalized Activation Function

We found the innovation while representing Equation (10) using the sigmoid function $S(x) = \frac{1}{1+e^{-x}}$ as follows:

$$\begin{aligned}
\text{Æ}(x^{(i)}) &= x^{(i)}S\big(-2\alpha(x^{(i)} - \mu_X - z_k\sigma_X)\big) \\
&= x^{(i)}S(ax^{(i)} + b))
\end{aligned} \tag{11}$$

In a previous work, Ramachandran et al. (2017) introduced the leverage of automatic search techniques to discover the best performance activation function. The experiments empirically discovered that the Swish activation function is the best performance activation function, defined as $xS(x)$ (Ramachandran et al., 2017). Intuitively, the definition of the Swish activation function is the same with Equation (11), and Equation (11) represents more generalized formula. Therefore, ASH (Adaptive SwisH) activation function provides the theoretical explanations for why Swish was the best performance activation function in the empirical evaluations. Therefore, we obtain the following.

**Lemma 3.** ASH activation function exhibits general formula for the Swish activation function.

Interestingly, the activation function designed for stochastic adaptive sampling is converged to the generalized Swish activation function. The extreme impression is that the stochastic percentile sampling by the activation function that mimics real neurons expresses the general formula of the swish activation function formerly known as state-of-the-art. Therefore, the stochastic percentile

sampling can partially be applied to the Swish activation function. Additionally, it can be supposed that the Swish activation function achieved superior performance in the previous studies based on the utilization of stochastic percentile sampling.

This paper initially considered an activation function that enables stochastic percentile sampling in a mathematically effective manner. However, we found that the mathematical expression of ASH supports the theoretical background of the Swish activation function. Therefore, this paper provides the theoretical backgrounds and rationales for the Swish activation function, which was empirically investigated. It is a significant innovation to provide the mathematical theorem that the Swish activation function is derived from a stochastically designed activation function.

## 3 Main Result

Similar to a previous study (Ramachandran et al., 2017), we compared ASH to several baseline activation functions on various models for different tasks using public datasets. Because many activation functions have been developed, we employed some of the most commonly used activation functions, namely ReLU, leaky ReLU (LReLU) (Maas et al., 2013), parametric ReLU (PReLU) (He et al., 2015), Softplus (Nair, Hinton, 2010), ELU (Clevert et al., 2015), SELU (Klambauer et al., 2017), and GELU (Hendrycks, Gimpel, 2016). In our experiments, every hyper-arameter in ASH and the other activation functions was set to be the same to demonstrate the advantages of ASH compared to other activation functions. In the tables, the highest accuracy values are highlighted in **bold**.

### 3.1 Classification Task

We first compared ASH to all the baseline activation functions on the CIFAR-10 (Krizhevsky et al., 2009), CIFAR-100 (Krizhevsky et al., 2009) datasets, and ImageNet (Russakovsky et al., 2015) datasets. We employed environments from a previous study (Ramachandran et al., 2017) and re-implemented the baseline models of ResNet-164 (He et al., 2016), wide ResNet28-10 (Zagoruyko, Komodakis, 2016), and DenseNet-100-12 (Huang et al., 2017). Based on these different environments, small differences were reported previously, but we believe that the accuracy trends are similar. We first evaluated ASH activation function against other activation functions using the ImageNet 2012 classification dataset because ImageNet is a widely utilized dataset in classification tasks. We then evaluated all activation functions using the CIFAR-10 and CIFAR-100 datasets, which have been widely utilized as benchmarks.

| Model | Top-1 Acc. (%) | | | Top-5 Acc. (%) | | |
|---|---|---|---|---|---|---|
| ReLU | $76.4 \pm 0.09$ | $75.6 \pm 0.10$ | $77.1 \pm 0.11$ | $91.2 \pm 0.09$ | $90.7 \pm 0.06$ | $90.7 \pm 0.06$ |
| LReLU | $77.6 \pm 0.07$ | $78.0 \pm 0.03$ | $76.6 \pm 0.07$ | $91.6 \pm 0.10$ | $91.2 \pm 0.07$ | $92.3 \pm 0.07$ |
| PLeLU | $77.0 \pm 0.13$ | $78.7 \pm 0.03$ | $78.0 \pm 0.09$ | $92.9 \pm 0.03$ | $92.3 \pm 0.14$ | $92.2 \pm 0.12$ |
| Softplus | $76.8 \pm 0.11$ | $77.3 \pm 0.03$ | $76.0 \pm 0.05$ | $91.5 \pm 0.12$ | $93.7 \pm 0.05$ | $93.8 \pm 0.11$ |
| ELU | $71.6 \pm 0.09$ | $73.7 \pm 0.09$ | $74.9 \pm 0.10$ | $85.6 \pm 0.06$ | $89.8 \pm 0.13$ | $90.2 \pm 0.08$ |
| SELU | $75.4 \pm 0.13$ | $78.1 \pm 0.14$ | $76.8 \pm 0.06$ | $91.6 \pm 0.09$ | $93.5 \pm 0.10$ | $90.9 \pm 0.04$ |
| GELU | $76.8 \pm 0.13$ | $77.9 \pm 0.05$ | $78.0 \pm 0.12$ | $90.2 \pm 0.11$ | $92.6 \pm 0.04$ | $91.9 \pm 0.09$ |
| Swish | $77.5 \pm 0.07$ | $76.6 \pm 0.06$ | $76.5 \pm 0.05$ | $92.2 \pm 0.12$ | $90.9 \pm 0.07$ | $92.2 \pm 0.07$ |
| ASH | $\mathbf{78.5 \pm 0.06}$ | $\mathbf{78.6 \pm 0.07}$ | $\mathbf{78.7 \pm 0.10}$ | $\mathbf{94.0 \pm 0.08}$ | $\mathbf{94.7 \pm 0.07}$ | $\mathbf{94.1 \pm 0.08}$ |

Table 1. ImageNet dataset. Three models are averaged. The values are mean and 95% confidence Interval (C.I.)

| Model | ResNet | WRN | DenseNet |
|---|---|---|---|
| ReLU | $94.4 \pm 0.04$ | $95.6 \pm 0.03$ | $95.7 \pm 0.02$ |
| LReLU | $94.5 \pm 0.05$ | $95.6 \pm 0.04$ | $94.7 \pm 0.09$ |
| PLeLU | $94.7 \pm 0.08$ | $95.4 \pm 0.03$ | $95.1 \pm 0.08$ |
| Softplus | $94.3 \pm 0.10$ | $94.2 \pm 0.08$ | $95.2 \pm 0.07$ |
| ELU | $93.5 \pm 0.10$ | $93.8 \pm 0.09$ | $94.5 \pm 0.11$ |
| SELU | $94.5 \pm 0.05$ | $95.8 \pm 0.07$ | $94.9 \pm 0.10$ |
| GELU | $95.2 \pm 0.04$ | $95.7 \pm 0.06$ | $94.8 \pm 0.10$ |
| Swish | $95.5 \pm 0.09$ | $95.6 \pm 0.08$ | $95.2 \pm 0.03$ |
| ASH | $\mathbf{95.7 \pm 0.08}$ | $\mathbf{96.7 \pm 0.04}$ | $\mathbf{96.0 \pm 0.11}$ |

Table 2. CIFAR-10 with mean values and 95% C.I.

| Model | ResNet | WRN | DenseNet |
|---|---|---|---|
| ReLU | $74.5 \pm 0.10$ | $78.4 \pm 0.04$ | $84.0 \pm 0.09$ |
| LReLU | $75.3 \pm 0.06$ | $77.9 \pm 0.07$ | $82.2 \pm 0.07$ |
| PLeLU | $74.7 \pm 0.06$ | $77.6 \pm 0.06$ | $82.1 \pm 0.07$ |
| Softplus | $76.1 \pm 0.05$ | $78.6 \pm 0.06$ | $84.1 \pm 0.02$ |
| ELU | $75.0 \pm 0.08$ | $76.4 \pm 0.09$ | $80.8 \pm 0.05$ |
| SELU | $73.4 \pm 0.05$ | $74.4 \pm 0.08$ | $81.4 \pm 0.06$ |
| GELU | $75.0 \pm 0.05$ | $78.2 \pm 0.10$ | $84.0 \pm 0.02$ |
| Swish | $75.7 \pm 0.10$ | $78.9 \pm 0.05$ | $84.0 \pm 0.03$ |
| ASH | $\mathbf{76.5 \pm 0.08}$ | $\mathbf{79.2 \pm 0.06}$ | $\mathbf{84.6 \pm 0.06}$ |

Table 3. CIFAR-100 with mean values and 95% C.I.

The ImageNet dataset evaluations were averaged based on the accuracy values of the three deep learning models. The results in Tables 1 to 3 highlight the outstanding performance of ASH activation function in terms of improving predictive accuracy. Because deep learning models for classification tasks demand sparsity, it is intuitive that ASH activation function improves accuracy compared to other activation functions.

## 3.2 Detection Task

We compared ASH to all the baseline activation functions on the COCO (Lin et al., 2014) and PASCAL VOC (Everingham et al., 2010) datasets for the detection task. Both of these datasets are widely used as benchmarks for detection tasks. We employed the same environments as the classification task and implemented the baseline models of Mask-R-CNN (He et al., 2017), SSD (Liu et al., 2016), and YOLOv4 (Bochkovskiy et al., 2020). For detection tasks, deep learning models output bounding boxes representing the locations of target objects. We exploited mAP@50 as an evaluation metric based on its popularity for detection tasks.

| Model | MR-CNN | SSD | YOLOv5 |
|---|---|---|---|
| ReLU | $68.5 \pm 0.02$ | $70.1 \pm 0.07$ | $72.1 \pm 0.04$ |
| LReLU | $68.9 \pm 0.06$ | $70.6 \pm 0.10$ | $72.6 \pm 0.03$ |
| PLeLU | $69.4 \pm 0.10$ | $71.0 \pm 0.11$ | $73.1 \pm 0.11$ |
| Softplus | $69.4 \pm 0.03$ | $71.0 \pm 0.08$ | $73.0 \pm 0.09$ |
| ELU | $69.4 \pm 0.10$ | $71.1 \pm 0.10$ | $73.0 \pm 0.06$ |
| SELU | $69.7 \pm 0.06$ | $71.2 \pm 0.04$ | $73.3 \pm 0.10$ |
| GELU | $70.0 \pm 0.08$ | $71.6 \pm 0.05$ | $73.7 \pm 0.04$ |
| Swish | $70.4 \pm 0.08$ | $72.0 \pm 0.03$ | $74.0 \pm 0.08$ |
| ASH | $\mathbf{71.1 \pm 0.06}$ | $\mathbf{72.7 \pm 0.02}$ | $\mathbf{74.8 \pm 0.09}$ |

Table 4. COCO with mean values and 95% C.I.

| Model | MR-CNN | SSD | YOLOv5 |
|---|---|---|---|
| ReLU | $65.8 \pm 0.03$ | $67.3 \pm 0.07$ | $69.3 \pm 0.09$ |
| LReLU | $66.9 \pm 0.07$ | $68.4 \pm 0.07$ | $70.5 \pm 0.07$ |
| PLeLU | $67.8 \pm 0.03$ | $69.2 \pm 0.07$ | $71.2 \pm 0.09$ |
| Softplus | $67.9 \pm 0.05$ | $69.4 \pm 0.03$ | $71.4 \pm 0.03$ |
| ELU | $67.8 \pm 0.05$ | $69.3 \pm 0.06$ | $71.4 \pm 0.06$ |
| SELU | $68.4 \pm 0.07$ | $69.9 \pm 0.03$ | $72.1 \pm 0.11$ |
| GELU | $68.9 \pm 0.07$ | $70.5 \pm 0.03$ | $72.4 \pm 0.11$ |
| Swish | $69.0 \pm 0.04$ | $70.6 \pm 0.02$ | $72.6 \pm 0.08$ |
| ASH | $\mathbf{70.5 \pm 0.06}$ | $\mathbf{72.1 \pm 0.04}$ | $\mathbf{74.1 \pm 0.11}$ |

Table 5. PASCAL VOC with mean values and 95% C.I.

The quantitative results in Tables 4 and 5 highlight the outstanding performance of ASH activation function compared to other activation functions. A higher mAP indicates that the predicted bounding boxes are closer to the annotations. ASH activation function provides superior performance for detecting target objects in various datasets for various deep learning models. Because the deep learning models used for detection tasks demand locality to generate bounding boxes, it is expected that $z_k$ will be small, demonstrating that greater activation can be realized using ASH activation function compared to the models used for the classification task.

## 3.3 Segmentation Task

We compared ASH to all of the baseline activation functions on the ADE20K (Zhou et al., 2017) and PASCAL VOC (Everingham et al., 2010) datasets for the segmentation task. Both datasets include many target objects in one scene. Therefore, they are widely utilized as benchmarks for segmentation tasks. We employed the same environments as the classification and detection tasks and implemented the baseline models of U-Net (Ronneberger et al., 2015), DeepLabV3+(DLV3+) (Chen et al., 2018), and EfficientNet (Tan, Le, 2019). Similar to other general benchmarks, we adopted intersection over union (IoU) and mean IoU (mIoU) values as evaluation metrics based on their popularity for segmentation tasks.

Similar to the previous tasks, the quantitative results in Tables 6 and 7 highlight the outstanding performance of ASH activation function compared to the other activation functions. Because locality is important for segmenting target objects from the background in segmentation tasks, it is intuitive that ASH activation function improves locality during feature extraction. The experimental results demonstrate that superior segmentation performance can be realized by using ASH activation function, which aids significantly in localizing target objects.

| Model | U-Net | DLV3+ | EfficientNet |
|---|---|---|---|
| ReLU | $49.4 \pm 0.04$ | $50.7 \pm 0.06$ | $52.2 \pm 0.03$ |
| LReLU | $49.8 \pm 0.07$ | $51.0 \pm 0.09$ | $52.3 \pm 0.05$ |
| PLeLU | $49.9 \pm 0.02$ | $51.0 \pm 0.03$ | $52.5 \pm 0.06$ |
| Softplus | $50.1 \pm 0.11$ | $51.2 \pm 0.10$ | $52.8 \pm 0.03$ |
| ELU | $50.3 \pm 0.04$ | $51.4 \pm 0.09$ | $52.8 \pm 0.08$ |
| SELU | $50.9 \pm 0.04$ | $52.1 \pm 0.03$ | $53.5 \pm 0.06$ |
| GELU | $50.9 \pm 0.07$ | $52.2 \pm 0.08$ | $53.5 \pm 0.05$ |
| Swish | $51.3 \pm 0.05$ | $52.4 \pm 0.02$ | $53.9 \pm 0.07$ |
| ASH | $\mathbf{53.4 \pm 0.05}$ | $\mathbf{54.7 \pm 0.08}$ | $\mathbf{56.3 \pm 0.09}$ |

Table 6. ADE20K with mean values and 95% C.I.

| Model | U-Net | DLV3+ | EfficientNet |
|---|---|---|---|
| ReLU | $74.3 \pm 0.05$ | $76.0 \pm 0.08$ | $78.1 \pm 0.07$ |
| LReLU | $76.2 \pm 0.06$ | $78.0 \pm 0.05$ | $80.3 \pm 0.10$ |
| PLeLU | $77.0 \pm 0.07$ | $78.9 \pm 0.06$ | $81.0 \pm 0.08$ |
| Softplus | $77.1 \pm 0.09$ | $78.8 \pm 0.03$ | $81.2 \pm 0.10$ |
| ELU | $77.2 \pm 0.10$ | $79.0 \pm 0.05$ | $81.3 \pm 0.02$ |
| SELU | $78.2 \pm 0.06$ | $80.1 \pm 0.03$ | $82.4 \pm 0.03$ |
| GELU | $78.9 \pm 0.06$ | $80.7 \pm 0.05$ | $83.2 \pm 0.08$ |
| Swish | $78.8 \pm 0.11$ | $80.7 \pm 0.04$ | $82.9 \pm 0.04$ |
| ASH | $\mathbf{81.2 \pm 0.04}$ | $\mathbf{83.2 \pm 0.04}$ | $\mathbf{85.5 \pm 0.05}$ |

Table 7. PASCAL VOC with mean values and 95% C.I.

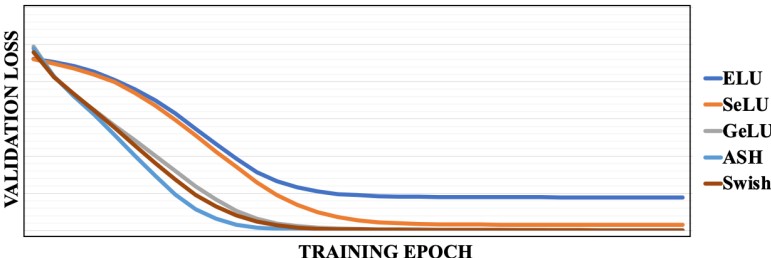

Figure 3: Validation loss values alongside the training epoch for the activation functions. The validation losses are averaged from the results of all experiments.

### 3.4 Training Time

We empirically explored the effectiveness of ASH activation function in terms of training time by monitoring the validation loss values for all activation functions. Fig. 3 reveals that the loss values of ASH activation function exhibit a steeper slope than those of the other activation functions. Therefore, because ASH activation function reaches convergence significantly faster than the other activation functions, we can empirically conclude that ASH has a superior effect in terms of reducing training time.

Through our experiments, we explored the outstanding performance of ASH activation function compared to other activation functions, including improvements in accuracy, sparsity, training time, and localization. In this study, the experimental results demonstrate the outstanding performance of ASH activation function. Additionally, supporting experiments and the results of other tasks such as image generation is presented in the *Supplementary Material*.

## 4 Conclusions

In this paper, we proposed a novel activation function to rectify inputs using an adaptive threshold considering the entire contexts of inputs more like human neurons. To this end, we designed an activation function to extract elements in the top-$k$ percentile from the input feature-map. Since sorting algorithm-based selections or quick selection algorithm demands a heavy computational cost, we employed the stochastic technique utilizing normal distribution to realize stochastic percentile sampling. Based on the mathematical derivations, we implemented ASH activation function in simple yet effective formula ($f(x) = x \cdot \text{sigmoid}(ax+b)$) with low computational cost for sampling the top-$k$ percentile from the input. In addition, we implemented ASH activation function, realizing (1) the adaptive threshold by employing the Z-score-based trainable variables and (2) the perception of entire contexts in rectifying an input by utilizing the mean and standard deviation of the input. Meanwhile, ASH activation function represented the generalized form of the Swish activation function that was empirically searched in the previous study. Therefore, this study also exhibited a novel contribution of the mathematical proofs for the state-of-the-art performance of the Swish activation function. Experiments using various deep learning models on different tasks (classification, detection, and segmentation) demonstrated superior performance for ASH activation function, in terms of accuracy, localization, and training time.

## Acknowledgment

This work was supported in part by the National Research Foundation of Korea (NRF) under Grant NRF-2020R1A2B5B01002786 and in part by the Bio & Medical Technology Development Program of the National Research Foundation (NRF) funded by the Korean government (MSIT) (No.2017M3A9G8084463).

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
