# Appendix. Stochastic Adaptive Activation Function

**Kyungsu Lee**
DGIST*
42988 Daegu, South Korea
ks_lee@dgist.ac.kr

**Jaeseung Yang**
DGIST
42988 Daegu, South Korea
yjs6813@dgist.ac.kr

**Haeyun Lee**
DGIST, SAMSUNG SDI
17084 Yong-In, South Korea
haeyun.lee@samsung.com

**Jae Youn Hwang**[†]
DGIST
42988 Daegu, South Korea
jyhwang@dgist.ac.kr

## Appendix A. Environment Description

The server included two CPUs of Intel(R) Xeon(R) Gold 6226R CPU @ 2.90GHz, 128GB RAMs, and eight Titan-Xp GPUs. Besides, we developed a deep learning models and activation functions using Tensorflow version 1 (Abadi et al., 2016) for the precise implementation. For the training, the batch size (Bottou, 2010) of the training was set to 32, and the Adam optimizer was utilized with the default values of all parameters (Kingma, Ba, 2014).

## Appendix B. Properties of ASH

ASH activation function is formulated as the following:

$$
\begin{aligned}
Æ(x^{(i)}) &= x^{(i)} S\big( -2\alpha(x^{(i)} - \mu_X - z_k \sigma_X)\big) \\
&= \begin{cases} x^{(i)} & \text{if } x^{(i)} \geq \mu_X + z_k \sigma_X, \\ 0 & \text{otherwise} \end{cases}
\end{aligned} \tag{A1}
$$

where $x^{(i)}$ is an element in input feature map $X$, and $\mu_X$ and $\sigma_X$ is the mean and the standard deviations of all elements in $X$. $S$ indicates *sigmoid* function, and $z_k$ is the variable with regard to sampling the top-$k\%$ percentile from $X$. Intuitively, ASH activation function is the threshold-based activation function rectifying inputs, and we obtained the following properties:

**Property 1.** ASH activation function is parametric.

We represented ASH activation function to be arithmetic and trainable due to $z_k$ concerning sampling percentile, and thus ASH is trainable and parametric. Thus, ASH activation function could exhibit different thresholds concerning the location or depth in a network. ASH activation function in the early layer exhibits a small threshold (large percentile) to retain substantial information, whereas ASH in deeper layers exhibits a small comparative percentile to rectify futile information. This property improves the superior rectification of ASH in deep neural networks.

**Property 2.** ASH activation function provides output concerning the contexts of the input.

Since the threshold value ($\mu_X + z_k \sigma_X$) is concerning the distribution of input $X$, the threshold value could be further fine-tuned with regard to the inputs. Compared to other threshold-based activation

---

*Daegu Gyeongbuk Institute of Science and Technology
†corresponding author

36th Conference on Neural Information Processing Systems (NeurIPS 2022).

functions, ASH exploits an adaptive threshold value, and thus it exhibits superior accuracy regardless of the variations in datasets.

Due to the novel properties, ASH activation function exhibits an improvement in imitating human neurons. More like human neurons compared to other activation functions, ASH provides output regarding the contexts of an input feature-map, and ASH exhibits different threshold values regarding the location, depth, or the types of the connected layers. To summarize, ASH exhibits novelty in imitating human neurons in terms of the activation function.

# Appendix C. Training curves

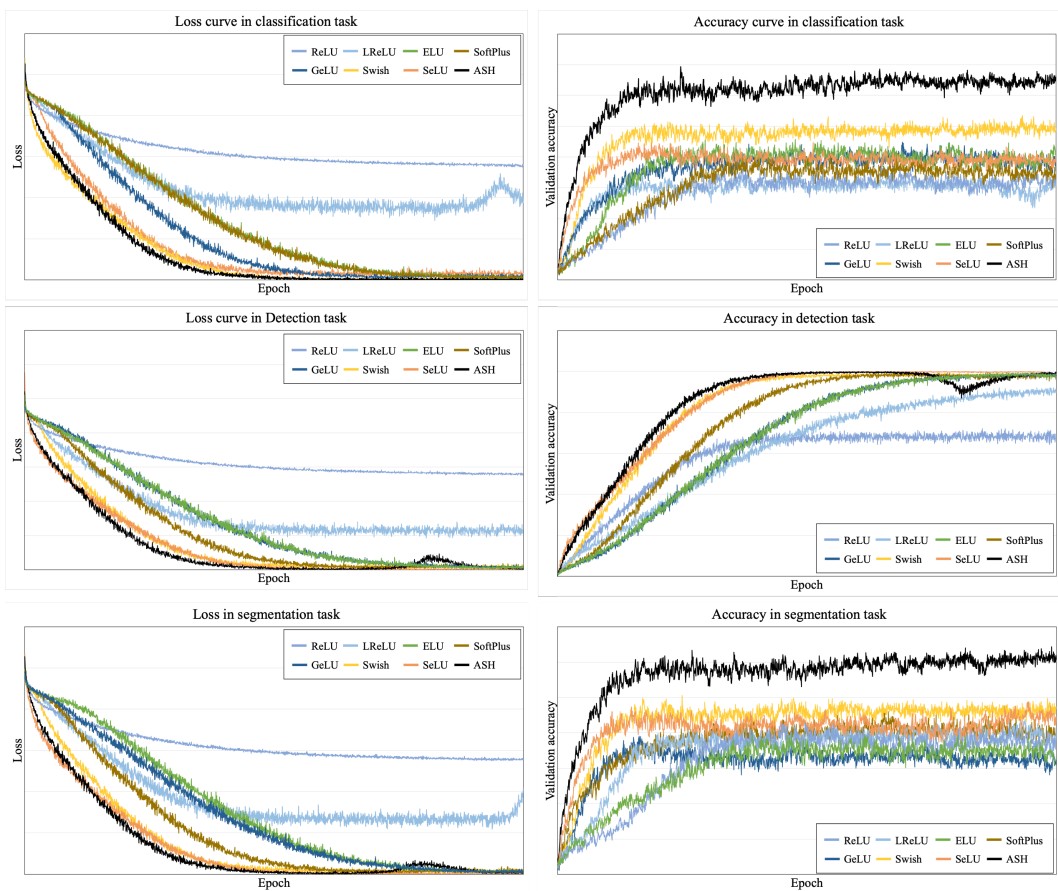

Supplementary Figure 1: Average training graph of ResNet-164,Wide ResNet28-10, and DenseNet-100-12 on ImageNet dataset using various activation functions along with ASH.

Supplementary Fig. 1 illustrates the training graph of loss values and validation accuracies. The experimental results demonstrate that ASH activation function is superior in training deep learning models for various tasks, including classification, detection, and segmentation. In particular, in the classification task, since ASHs in the early layers provide broad activation and ASHs at the end of the model rectify informative features (**Property 1**), ASH significantly improves the training efficiency and accuracy. Similarly, ASH exhibits significant localization properties like attention mechanism, and thus ASH achieved superior segmentation performance. On the other hand, ASH improves the predictive accuracy of the bounding boxes in the detection task, whereas it degrades the confidence score due to its localization property. Therefore, the accuracy of ASH activation function somewhat decreases at the end of the training. Here, the x-axis indicates the percentage of training epoch, and they were averaged. In addition, the y-axis indicates the range of (0, 0.8).

# Appendix D. Classification task

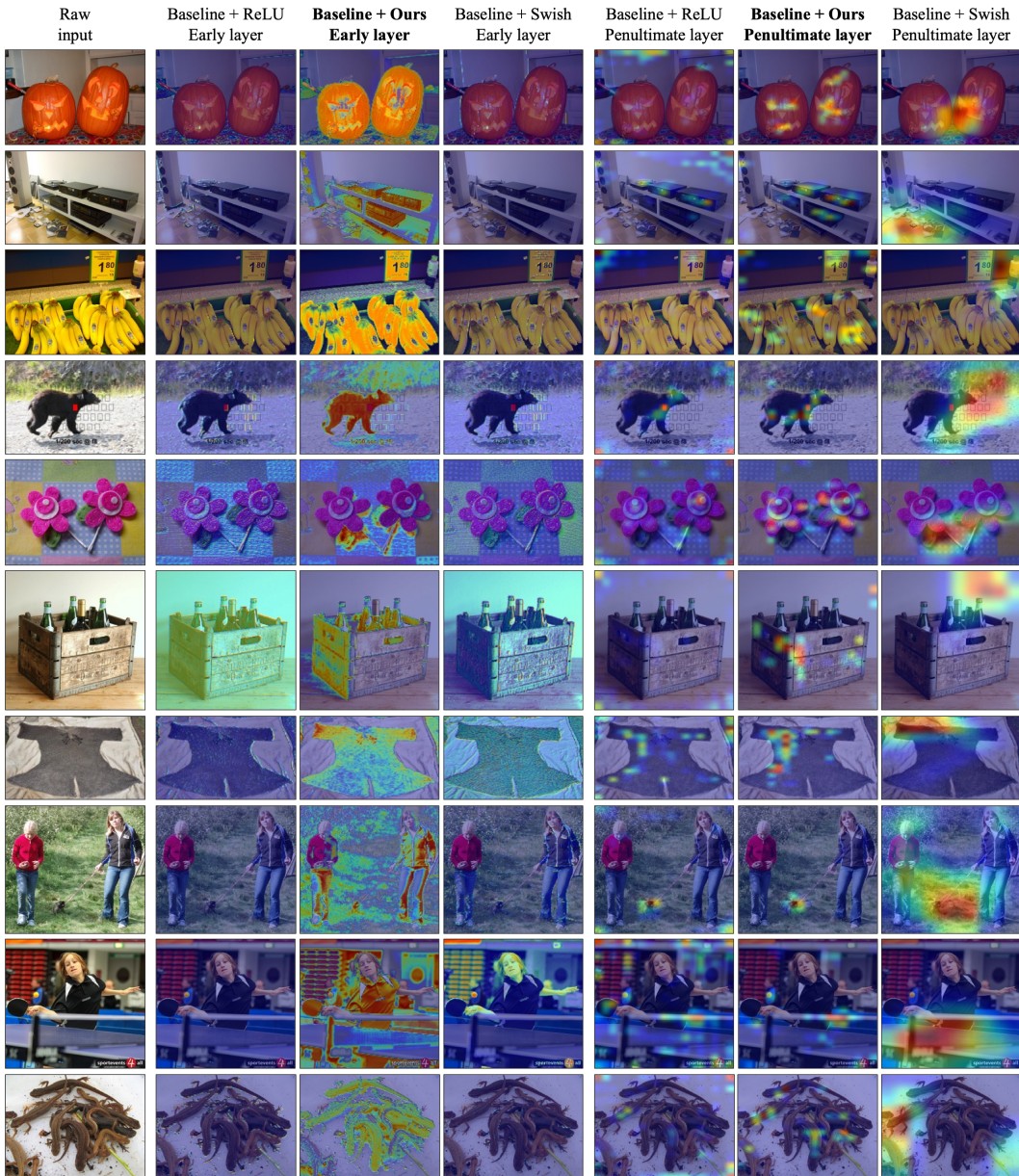

Supplementary Figure 2: Grad-CAM samples generated by Baseline models with ReLU, ASH, and Swish activation functions using Imagenet dataset. ResNet-164 (1∼5 rows) and Dense-Net (6∼10 rows) are used as baseline models.

Supplementary Fig. 2 illustrates the GRAD-CAM (Selvaraju et al., 2017) samples by using ResNet-164 and Dense-Net models with ReLU, Swish, and ASH activation function in the classification task of ImageNet dataset. Here, since ASH is based on the threshold-based activation function, ASH exhibits discrete activations like ReLU. In Supplementary Fig. 2 **Property 1** is clearly illustrated. In the early layer, ASH activation function provides sufficiently broad but informative activations with regard to the target objects to forward layers. Besides, ASH at the end of the models exhibits the activations that are discrete but localized onto the target object. Therefore, ASH activation function could provide informative activations to the deep learning models, and thus it could improve the superior accuracy in every task.

# Appendix E. Segmentation task

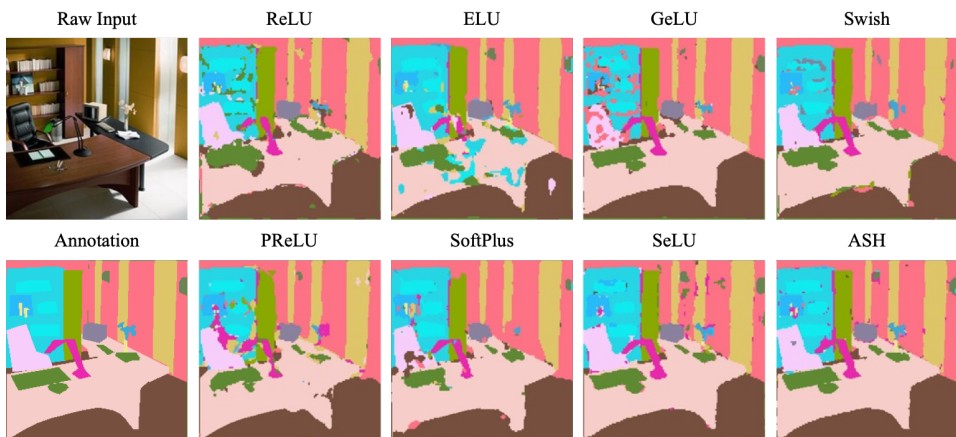

Supplementary Figure 3: Samples segmented by U-Net (Ronneberger et al., 2015) with ReLU, PReLU, ELU, SoftPlus, GELU, SeLU, Swish, and ASH activation functions using ADE20K dataset.

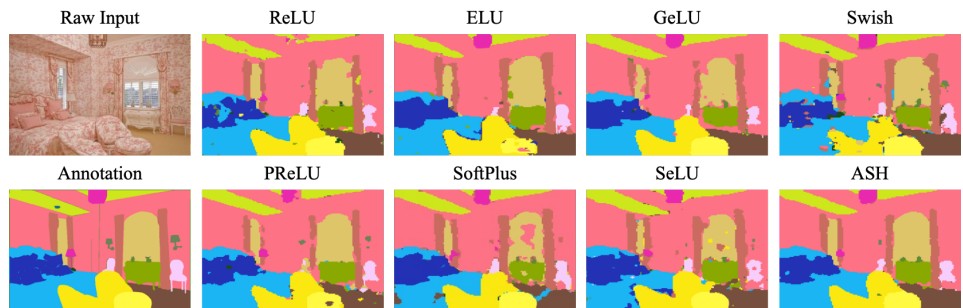

Supplementary Figure 4: Samples segmented by DeepLabV3+ (Chen et al., 2018) with ReLU, PReLU, ELU, SoftPlus, GELU, SeLU, Swish, and ASH activation functions using ADE20K dataset.

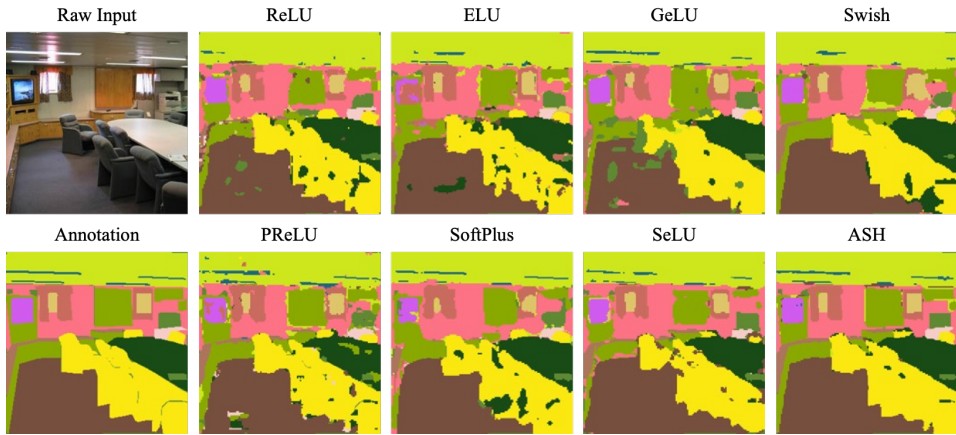

Supplementary Figure 5: Samples segmented by EfficientNet (Tan, Le, 2019) with ReLU, PReLU, ELU, SoftPlus, GELU, SeLU, Swish, and ASH activation functions using ADE20K dataset.

# Appendix F. Image generation task

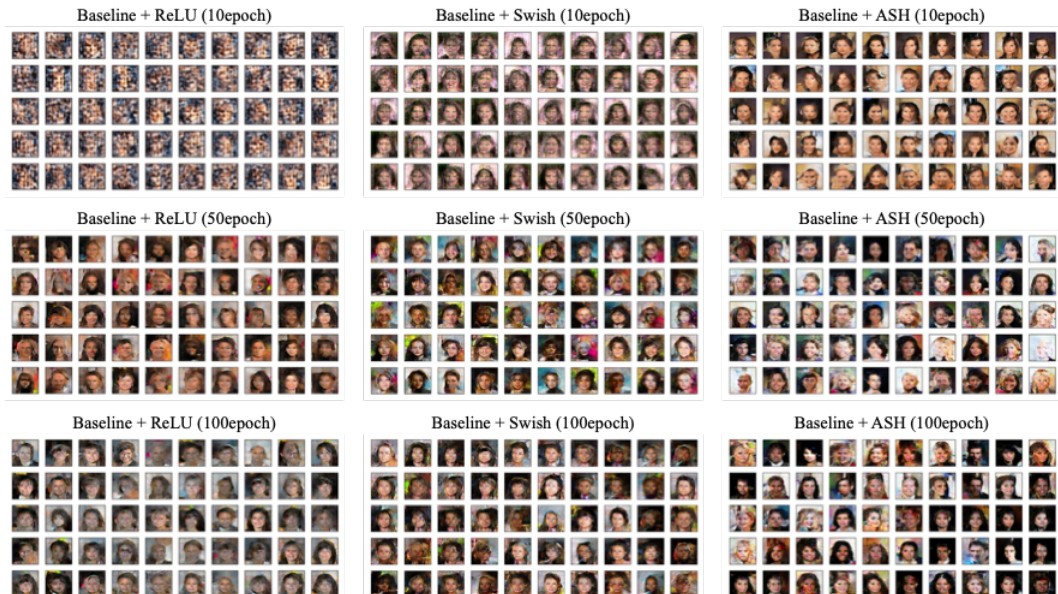

Supplementary Figure 6: Samples generated by DCGAN with ReLU, Swish, and ASH activation functions using celebA dataset.

Supplementary Fig. 6 illustrates the generated samples by DCGAN (Radford et al., 2015) with ReLU, Swish, and ASH activation functions using celebA dataset (Yang et al., 2015). Despite the similar quantitative results by every activation function, ASH activation function significantly reduces the training time. In particular, the generated images by the DCGAN model with ASH activation function are explicitly exhibited as more like human from the early epoch (10). The experiment also demonstrates that ASH activation function could significantly improve the training efficiency due to its advantages of (**Property 1**) and (**Property 2**).

# Appendix G. Formulation of ASH

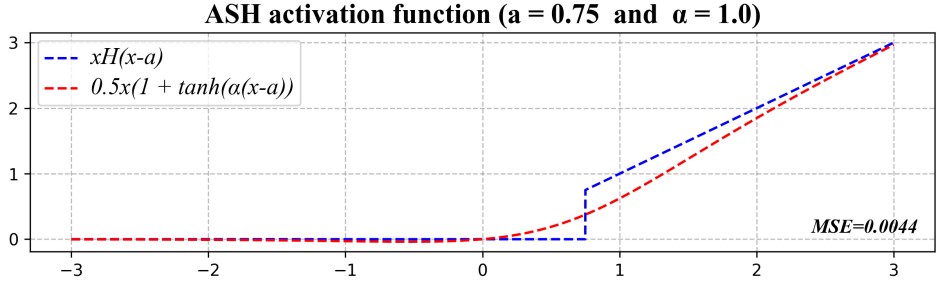

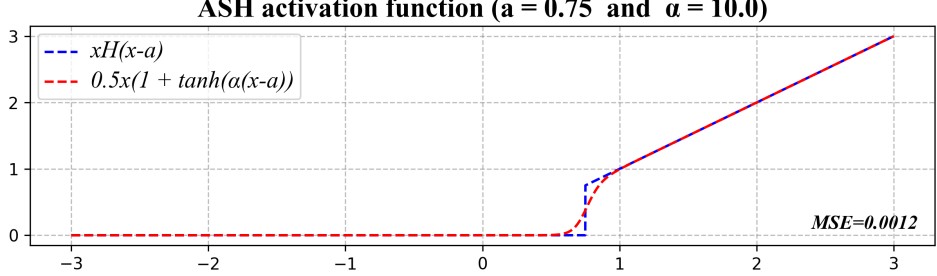

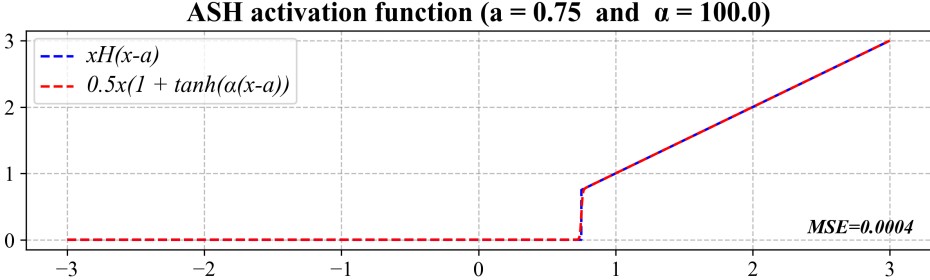

Supplementary Figure 7: Various versions of ASH activation functions with various values of $\alpha$ in Equation (10). As $\alpha$ increasing, two functions are approximated as similar.

In equation (10), ASH activation function is initially defined using Heaviside Step Function, and it is approximated using the sigmoid function as follows:

$$Æ(x^{(i)}) = x^{(i)} H(x^{(i)} - \mu_X - z_k \sigma_X) \tag{A2}$$

$$Æ(x^{(i)}) = \frac{1}{2}x^{(i)} + \frac{1}{2}x^{(i)} \tanh(\alpha(x^{(i)} - \mu_X - z_k \sigma_X)) \tag{A3}$$

$$Æ(x^{(i)}) = \frac{x^{(i)}}{1 + e^{-2\alpha(x^{(i)} - \mu_X - z_k \sigma_X)}} \tag{A4}$$

Here, Equations (A2) and (A3) exhibit the same equation since $\tanh(x) = \frac{e^x - e^{-x}}{e^x + e^{-x}}$. To simplify, Equation (A3) can be expressed using the substitution as the following:

$$f(x) = \frac{x}{1 + e^{-2\alpha x + \beta}} \tag{A5}$$

Then, suppose a large value of $\alpha$ and Equation (A5) is expressed as the follows:

$$f(x) = \lim_{\alpha \to \infty} \frac{x}{1 + e^{-2\alpha x + \beta}} \tag{A6}$$

To clarify, we consider two cases of (1) $x \geq 0$, and (2) $x < 0$. In the first case, $e^{-2\alpha x + \beta}$ is converged to 0, and thus $f(x) = x$. In contrast, in the second case, $e^{-2\alpha x + \beta}$ is diverged, and thus $f(x) = 0$. Therefore, $f(x)$ is approximated as $max(0, x)$. Here, since the definition of ASH activation function is originally as below, ASH could be approximated as the following:

$$\cancel{E}(x^{(i)}) = \begin{cases} x^{(i)} & \text{if } x^{(i)} - \mu_X - z_k \sigma_X \geq 0, \\ 0 & \text{otherwise} \end{cases}$$

$$= max(0, x^{(i)} - \mu_X - z_k \sigma_X) \tag{A7}$$

$$= \frac{x^{(i)}}{1 + e^{-2\alpha(x^{(i)} - \mu_X - z_k \sigma_X)}}$$

Supplementary Fig. 7 shows the various versions of ASH alongside the various values of $\alpha$. As the value of $\alpha$ increasing, ASH activation function ($xH(x-a)$) could be reasonable approximated as hyperbolic function ($0.5x(1 + \tanh(\alpha(x-a)))$).

## Appendix H. Comparison of ASHs

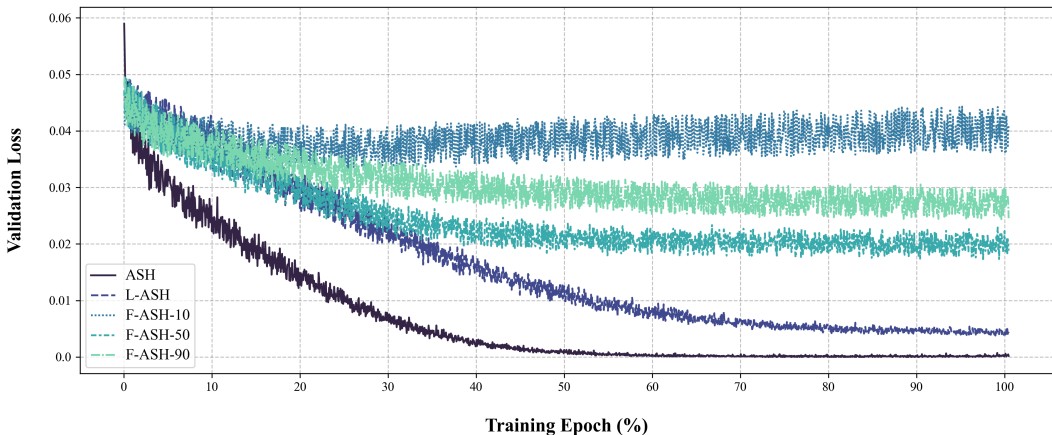

Supplementary Figure 8: Comparison of various versions of ASH activation function. ASH activation function indicates the proposed activation function, L-ASH indicates Leaky ASH, and F-ASH-$k$ is ASH activation function that $k$-precentile is fixed rather than trainable.

ASH activation function that rectified top-$k\%$ percentile could be modified into various versions. Suppose Leaky ASH (L-ASH) that utilizes the scaling factor like Leaky ReLU, such that L-ASH is defined as follows:

$$\mathcal{A}(x^{(i)}) = \begin{cases} x^{(i)} & \text{if } x^{(i)} \text{ is ranked in top } k\% \text{ percentile of } X, \\ ax & \text{otherwise} \end{cases} \tag{A8}$$

Here, $a$ is trainable like Leaky ReLU and positive value. L-ASH could improve the gradient vanishing problems. Additionally, we can employ fixed $k\%$ percentile in ASH, and we denoted it as Fixed ASH with $k$ (F-ASH-$k$). F-ASH-$k$ rectifies the top-$k$ percentile from inputs but $k$ is fixed as a hyper-parameter. To find the best performance ASH, we compared the various version of ASH

activation functions; ASH, L-ASH, F-ASH-10, F-ASH-50, and F-ASH-90. Supplementary Fig. 8 shows the validation loss value alongside the training epoch for activation functions. The validation losses are averaged from the results of all experiments of classification, detection, and segmentation.

The experimental result illustrates that the proposed ASH activation function exhibits early convergence with lower validation loss values. Compared to ASH activation function, L-ASH exhibits slower convergence for the optimization due to the increased number of trainable parameters and decreased sparsity caused by $a$ in the negative domain below top-$k$% percentile. Furthermore, F-ASH-$k$ activation functions exhibit lower performance in terms of validation loss since they rectify the fixed percentile of inputs regardless of contextual information of inputs. Here, F-ASH-50 shows slow but continuous convergence to the optimization since 50% rectification could be regarded as ReLU activation function. In contrast, F-ASH-90 has limitations in the optimization since it could not localize the informative features in the inputs. Similarly, F-ASH-10 has limitations in the optimization since it extremely rectifies the overall information of inputs. The results conclude the benefits of ASH activation functions in terms of sparsity and trainable property.

## Appendix I. Discussion

**Wide Localization (deep depth)**          **Narrow Localization (shallow)**

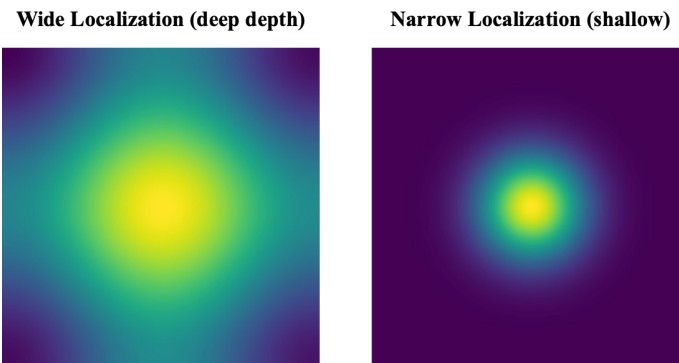

Supplementary Figure 9: Localization property of ASH. Localization and pass out are realized in the wide fields in ASH activation function in deeper depth. Narrow localization and extreme rectification are realized in ASH activation function in shallow depth.

**Localization property of ASH**       Supplementary Fig. 9 illustrates the localization example of ASH activation function. $z_k$ of ASH in a deeper depth exhibited smaller values (even negative values) compared to $z_k'$ of another ASH in a shallower depth. Note that, a small value of $z_k$ implies the wide range of top-$k$% percentile, and thus it leads to the wide field of localization. In contrast, a large value of $z_k$ implies the narrow range of top-$k$% percentile, and thus it provides extreme rectification and a narrow field of localization. Again, in the early layer, ASH activation function provides sufficiently broad but informative activations to forward layers. Additionally, ASH at the end of the models exhibits the activations that focus on the target object. Therefore, ASH activation function could provide informative activations to the deep learning models. Note that, the deep depth layers indicate that the layers are close to the input layer, whereas the shallow depth layers indicate that the layers are close to the output layers.

**Parameter Selection of ASH**       In this study, we proposed ASH activation function of which parameter $z_k$ is trainable and regarded as the hyper-parameter. However, ASHs exhibit significantly different $z_k$ values with significant variations even in the same networks. In addition, different values of $z_k$ are utilized concerning different tasks. Intuitively, a wide range of activations is required to recognize entire contexts of an image in a deeper depth, whereas a narrow range of activations is required to rectify features in a shallower depth in image recognition, especially a classification task. In contrast, consistent activation of the target object is required, in the segmentation or detection tasks. For instance, ASH in the initial position of the network exploits small $z_k$ (e.g., 90% percentile sampling), whereas ASH at the end of the network exhibits an enormous $z_k$ value (e.g., 15% percentile sampling). Therefore, rather than searching for the best performing parameters, we assumed that learning naturally from the network itself imitates more like human neurons. Thus, we skipped exploring the best parameter for ASH, in terms of $z_k$.

**Future Work** Despite the experimental results demonstrating the superior performance of ASH activation function in accuracy, sparsity, training time, and localization property, the mathematical proofs of those properties are limited. The supporting mathematical analysis and proofs could be more discussed as potential future work. Furthermore, we mainly applied ASH activation functions to the task in which the role of the activation function is significantly issued, including classification, detection, segmentation, and image generation task. Since vision-based analysis could exhibit the properties of activation functions in a visual manner alongside the mathematical proofs, we mainly utilized the vision-based tasks. Text- and transformer-based analysis could be further discussed for future work. Furthermore, ASH activation function could be applied in the field of Natural Language Processing or Signal Compression field rather than in vision-based applications. This also remains as future work.