# OpenReview forum: "Stochastic Adaptive Activation Function"
_NeurIPS.cc/2022/Conference — NeurIPS 2022 Accept_

### Official Review · Reviewer_ecwn · 2022-06-17

**Rating:** 3
**Confidence:** 5
**Soundness:** 2 fair
**Presentation:** 1 poor
**Contribution:** 2 fair

**Summary:**

This paper proposes an rectifier based activation function where the threshold is set to the upper k% of the input tensor values.

**Questions:**

- How is k determined?  Is it fixed, or learnable?  Is it the same across a network, or different for each layer?  What values for k are used in the experiments?

- The approximation of heaviside step function relies on a large value of alpha.  How does this affect the analysis in Equation 10?

**Limitations:**

- Some claims are excessive and unsubstantiated.  For example: "our activation function exhibits outstanding performance in any deep learning applications," or "the experiments empirically discovered that the Swish activation function is the best performance activation function."

- The experiments do not show any case in which ASH is outperformed by any other function.  The implication is that it is universally the best activation function.  This implied claim is problematic because there are no repeated runs (with the exception of 3 runs in Table 1) nor any statistical significance tests to back up these results.  See checklist item 3c.

- Figure 3 has no numbers on the x or y axes.

**Strengths And Weaknesses:**

- Originality: The main originality in this paper comes from determining a threshold based on a percentile from the input entries.  To my knowledge, this has not been considered before.

- Quality: The main issue with the paper is that there are no statistical significance tests for the final results.  This prevents drawing any substantial conclusions from the results, which hurts the paper's quality.

- Clarity: The writing is at times difficult to follow and understand, but the mathematics appear sound.

- Significance: The paper would be much more significant with repeated runs and statistical significance tests, but as-is, it is unsignificant.

---

> ### Author Response · Authors · 2022-08-02
> **Response to Reviewer ecwn**
>
> ____
> __(Question 1) How is k determined? Is it fixed, or learnable? Is it the same across a network, or different for each layer? What values for k are used in the experiments?__
>
> According to the Z-Table, $k$ and $z_k$ are associated in terms of Z-score as illustrated in Equations (4) and (5). In addition, $z_k$ is determined to be trainable (line 190-191). Therefore, $k$ is also trainable in ASH. Furthermore, $k$ ($z_k$) exhibits different values across a network (different values for each layer), as illustrated in Property 1 (Supplementary File Line 16-17) and Discussion (Supplementary File Page 5). In the training phase, all $k$s are initialized as 50% (0.5) and $z_k$s are as 0.0 based on Z-Table.
>
> ____
> __(Question 2) The approximation of Heaviside step function relies on a large value of alpha. How does this affect the analysis in Equation 10?__
>
> In Equation (10), the second line and the third line are mathematically the same. To approximate the Heaviside step function into a hyperbolic tangent or sigmoid function, those functions require a large coefficient in front of $x$. More detailed explanations are illustrated in Appendix G section (Supplementary File).
>
> ____
> __(Limitation 1) Some claims are excessive and unsubstantiated. For example: "our activation function exhibits outstanding performance in any deep learning applications," or "the experiments empirically discovered that the Swish activation function is the best performance activation function."__
>
> The novel properties of ASH activation function are well explained in Supplementary File (Appendix A-F) and experimental results. The main highlights of ASH activation function are (1) achieving high accuracy in various tasks of classification, detection, and segmentation [Experimental results], (2) adaptive attention mechanism [Appendix D and E], and (3) early convergence from the early epochs with lower loss values and high accuracy [Appendix C and F]. Additionally, we modified  Abstract and related paragraph in Main text to exhibit detailed properties of ASH activation function (high accuracy and early convergence), as well as the experimental results by adding statistical analysis.
>
> Furthermore, “the experiments empirically discovered that the Swish activation function is the best performance activation function.” is addressed in the original paper on the Swish activation function (Ramachandran et al., 2017), and we excerpted this to highlight the relations between Swish and ASH. To clarify, we added the reference in this sentence.
>
> ____
> __(Limitation 2) The experiments do not show any case in which ASH is outperformed by any other function. The implication is that it is universally the best activation function. This implied claim is problematic because there are no repeated runs (with the exception of 3 runs in Table 1) nor any statistical significance tests to back up these results. See checklist item 3c.__
>
> We modified the related Tables according to your and other reviewers’ thoughtful comments. We modified the manuscript by adding 95% Confidence Interval (C.I.) with mean values.
>
> ____
> __(Limitation 3) Figure 3 has no numbers on the x or y axes.__
>
> We modified Figure 3.
>
> ____

---

> > ### Comment · Reviewer_ecwn · 2022-08-03
> > **Response to author comments**
> >
> > I appreciate the authors taking the time to address my comments.  Although the tables now include confidence intervals, the number of samples is not mentioned. Also, the mean accuracy is identical in every entry to that in the original manuscript. It seems like the updated accuracies (which should have changed with repeated runs) were left out.
> >
> > These shortcomings are serious: it is impossible to gauge the significance of the paper because of them, and I significantly disagree with the reviewers awarding this paper high scores.

---

> > > ### Author Response · Authors · 2022-08-05
> > > **Response to Reviewer ecwn**
> > >
> > > Dear Reviewer ecwn.
> > >
> > > Thank you for your comments and for taking the time to review my paper.
> > >
> > > We agree that the statistical analysis and statistical values are crucial in the paper. However, the related information has been already shown in the manuscript (Lines 252, 254, 268, and 284).
> > >
> > > As mentioned in the manuscript, we conducted the experiments in environments that are as similar as possible to the previous paper [1]. In the previous paper, they tested the model with three different parameters three times. Hence, one model was tested nine times, and we showed the statistical values that are averaged from nine trials. Furthermore, we have already applied the repeated runs, but we only missed to exhibit standard deviation values in the tables of the previous manuscript. Hence we only updated the tables by adding standard deviation values without any changes in mean values.
> > >
> > > To clarify the manuscript, we will revise the manuscript or supplementary file by adding detailed environmental descriptions.
> > >
> > >
> > >
> > > Again, Thank you for your comments, and we agree that your comments have improved the quality of the manuscript.
> > >
> > >
> > > Thanks.
> > >
> > >
> > > [1] Ramachandran, Prajit, Barret Zoph, and Quoc V. Le. "Swish: a self-gated activation function." arXiv preprint arXiv:1710.05941 7.1 (2017): 5.

---

> > > > ### Comment · Reviewer_ecwn · 2022-08-05
> > > > **Response to authors**
> > > >
> > > > Thank you for clarifying, and I apologize for misunderstanding that the repeated runs were already conducted for the initial submission.
> > > >
> > > > Could you double check that the accuracies and confidence intervals are in the same scale?  For example, the accuracy is multiplied by 100 (0.748 becomes 74.8) and it is unclear whether the same was done to the confidence interval.  Often individual runs can vary by a percentage point, and at first glance the confidence intervals do seem a bit small.
> > > >
> > > > My main concern about the statistical significance of the results has been addressed.  In its current form, I would consider the paper a weak reject and increase my score from 3 to 4.
> > > >
> > > > I'm now satisfied with the technical results of the paper, but the writing itself would benefit from another round of improvement.  Experimental details are lacking, for example: number of runs for confidence intervals, Figure 3 is still missing axis values on the plot, the caption to Figure 3 doesn't mention which dataset, architecture, or hyperparameters lead to those loss curves, in Table 1 an ASH result is bolded even though it is outperformed by PLeLU, in Table 1 it is not clear which column corresponds to which architecture, and in general the writing is difficult to follow.
> > > >
> > > > If the writing issues were not present I would consider the paper a weak accept / score of 5.  There is obviously time to address them before publication but in my opinion they should not still be unresolved at this stage of reviewing.

---

> > > > > ### Author Response · Authors · 2022-08-07
> > > > > **Response to Reviewer ecwn**
> > > > >
> > > > > ____
> > > > > __(Question 1) Could you double check that the accuracies and confidence intervals are in the same scale? .... the confidence intervals do seem a bit small.__
> > > > >
> > > > > Yes, the scale of the resultant tables is all percentage (%) levels. We agree that 95% ci seems a bit small at first glance, but this is because of the large number of samples (N). Since 95% CI is calculated from $z\frac{\sigma}{\sqrt N}$, the scale of 95% CI is affected by the number of samples. If the number of samples is extremely large, then the 95% CI would be decreased at seemed.
> > > > > For instance, in Table 4, ASH activation function with MR-CNN achieved 71.1% (=0.711) accuracy and 0.06% (=0.0006) 95% CI. Here, those values were calculated from 900 samples in which average is 0.711, 0.00990183 standard deviation ($1.96 \times \frac{0.00990183}{\sqrt 900} = 0.0006$). Although the minimum value of the samples is 69.41% (=0.6941) and the maximum value of the sample is 72.79% (= 0.7279), 95% ci seems a bit small due to a large number of samples.
> > > > > To clarify the experimental environment, we will modify the manuscript as below (Question 2).
> > > > >
> > > > > ____
> > > > >
> > > > > __(Question 2) Experimental details are lacking, for example: number of runs for confidence intervals__
> > > > >
> > > > > We understand and agree that the experimental details are important including the number of trials and exhibiting the experimental environment. As we mentioned, the experiments are based on the previous paper [1], but we now agree that a detailed illustration of our own experimental environment would be better for the understanding of readers rather than referring to the previous studies. We will update the experimental environment in Appendix A (Supplementary File), as follows:
> > > > >
> > > > > *In the experiments, we basically employ the same environment as the previous study [1], including hyperparameters, trials, and repeated runs. In our experiments, all deep learning models have trained three different learning rates three times, and thus each deep learning model is trained nine times for each learning rate. Furthermore, nine times training were repeated by varying random initialization and random sampling ten times. Therefore, totally 900 samples are produced for each model. Tables in the main results are generated from the statistical analysis of those 900 samples, by indicating mean values and 95% confidence interval.*
> > > > >
> > > > > ____
> > > > >
> > > > >
> > > > > __(Question 3) Figure 3 is still missing axis values on the plot, the caption to Figure 3 doesn't mention which dataset, architecture, or hyperparameters lead to those loss curves__
> > > > >
> > > > > We previously updated information about Figure 3 in the caption. We hope you find information at the end of Caption 3 (Page 9). The X-axis indicates the percentage of the training process (0-100%), the Y-axis indicates the validation loss, and the range is (0, 0.8).
> > > > >
> > > > > ____
> > > > >
> > > > > __(Question 4) Figure 3 doesn't mention which dataset, architecture, or hyperparameters lead to those loss curves__
> > > > >
> > > > > As we illustrated in the main paper (in Caption 3), the graph was averaged from every trial, every hyperparameter, and every task, including classification, detection, and segmentation. We understand that simple averaging degrades the statistical analysis detail, but we illustrated Fig. 3 by averaging all trials since the previous works [1-3] related to activation functions have exhibited similar results. In addition, since the tendency of [training epoch]-[validation loss] has exhibited similar trends, we would illustrate the averaged graph to simplify the tendency from every task. The result demonstrates that the usage of ASH activation could provide early convergence compared to other activation functions, and this tendency has been observed in every task of classification, detection, segmentation, and even image generation.
> > > > >
> > > > > ____
> > > > >
> > > > > __(Question 5) In Table 1 an ASH result is bolded even though it is outperformed by PLeLU, in Table 1 it is not clear which column corresponds to which architecture, and in general the writing is difficult to follow.__
> > > > >
> > > > > Sorry for the missing detail, we will modify the incorrect table by re-bolding the PReLU result (78.7 ± 0.03).
> > > > >
> > > > > In addition, as the same with the previous question, (as mentioned Caption) the results of Table 1 are averaged from all trials, all hyperparameters, and all repeated runs of ResNest, WRN, and DenseNet. We illustrated Table 1 by averaging all trials like the previous works [1-3], but we understand that illustrating our experimental details are important, we will update the manuscript to the above questions.
> > > > >
> > > > > Furthermore, we will look over the overall manuscript carefully and correct all the grammatical errors, and we will further take the English Editing Service. Although our writing style and some errors in our manuscript would not be satisfactory for the reviewer,we believe that our manuscript could exhibit the technical novelty of ASH activation function through well-organized mathematical explanations and proofs.
> > > > >
> > > > > ____

---

### Official Review · Reviewer_aJG2 · 2022-07-02

**Rating:** 9
**Confidence:** 4
**Soundness:** 4 excellent
**Presentation:** 4 excellent
**Contribution:** 4 excellent

**Summary:**

The authors propose a new trainable activation function that unlike most activation functions that work element-wise, this one activates over complete layers.

More precisely, the activation function behaves like a ReLU, either activating with the input value, or cancelling to zero when deactivating. However, the activation criteria is whether the element is in the top k% of the elements in the layer.

The threshold is then controlled by a learnable parameter, making it a trainable activation function.

**Questions:**

The empirical evaluation shows that this activation function is valuable and it works.
I have however a series of questions that could be interesting to analyze to better understand the proposed method.

I'm thinking of the similarities of this activation with dropout. In particular, I'm wondering if the cancelation of the elements, should lead to a scale adjustment? It would seem like that could be beneficial as in dropout.

Now, considering the cancelation induced by the activation function, there are big parts of the network that won't receive gradient updates if they are not in the top k%. Would it make sense to introduce something like a negative tail or a scaling factor for the elements below the top k%? Think of Leaky ReLU.

Furthermore, as one would normally use these activation functions through the network. I'm wondering if there is some effect of vanishing gradients happening, or signal reduction, due to the cancellation imposed after each layer. I could imagine that if the learned parameter would cancel 90% of the signal at each layer, this could introduce a strong loss of information.

I'm also wondering whether this is a new approach for signal compression?. Consider a fixed top k% parameter. This directly implies that the number of outputs after the activation is a fraction of the size of the output. This is potentially equivalent to a reduction layer.
Here you do not get a dimensionality reduction, as for every input still different outputs could activate. But there is a compression in the amount of data reaching the next layer. If you were to control the percentage of information dropped, you could construct something like an auto-encoder and get the benefit of signal compression, without the effort needed in compressing the dimensionality.

**Limitations:**

no negative societal impact

**Strengths And Weaknesses:**

The paper is very interesting and I believe it could have applications and connections to other areas in deep learning.

The paper is well written and the empirical evaluation shows that the proposed activation works well.

The major weakness I see, is that the authors compare their activation mostly with other non-trainable activations. However, it is known that trainable activation functions such as [1], etc., usually perform better and such comparisons are missing in the empirical evaluation.

[1] Molina, Alejandro, Patrick Schramowski, and Kristian Kersting. "Padé Activation Units: End-to-end Learning of Flexible Activation Functions in Deep Networks." International Conference on Learning Representations. 2019.

---

> ### Author Response · Authors · 2022-08-02
> **Response to Reviewer aJG2**
>
> ____
> __(Weakness) The major weakness I see, is that the authors compare their activation mostly with other non-trainable activations. However, it is known that trainable activation functions such as [1], etc., usually perform better and such comparisons are missing in the empirical evaluation.__
>
> In the experiments, we compared ASH activation function to Leaky ReLU and Parametric ReLU which are trainable activation functions and commonly utilized in deep learning networks.
>
> However, thank you for introducing an interesting paper. We reviewed the paper, and we found that both activation functions exhibit the trainable property. However, ASH activation function exhibits a difference in recognizing the contextual information of inputs as well as adaptive thresholding properties. We added the reference to this paper, and we would like to examine quantitative analysis alongside ASH activation function in future work.
>
> ____
>
> __(Question 1) I'm thinking of the similarities of this activation with dropout. In particular, I'm wondering if the cancelation of the elements, should lead to a scale adjustment? It would seem like that could be beneficial as in dropout.__
>
> Yes, this study initially has started from the adaptive dropout methods, and thus ASH activation function exhibits a cancelation property. Therefore, ASH activation function could provide the benefit in terms of sparsity as a dropout [1]. Additionally, we commonly utilized normalization methodologies with ASH activation function in the deep learning applications, and the detailed architecture could be found in the GitHub codes after the publishment.
>
> ____
> __(Question 2) Now, considering the cancelation induced by the activation function, there are big parts of the network that won't receive gradient updates if they are not in the top k%. Would it make sense to introduce something like a negative tail or a scaling factor for the elements below the top k%? Think of Leaky ReLU.__
>
> As the reviewer commented, the variable could not be updated by the backpropagation when the corresponding output is not in the top-k\% percentile. This could happen in some samples. However, deep learning networks commonly employ a large number of images to train the networks, and thus it rarely happens that one variable could not be included in the top-k\% percentile in terms of data augmentation. We were impressed by the reviewers’ comments and they could have improved the quality of the manuscript.
>
> We agree that the scaling factor for the elements below the top-k\% percentile could be effective and we conducted the related experiments before the initial submission. The experiment showed that the Leaky ASH (L-ASH) exhibited slower convergence for the optimization due to the increased number of trainable parameters and decreased sparsity. During the experiments, we noticed that the sparsity of ASH activation function could provide superior benefits for the optimization of deep learning networks. As the reviewers commented, we added the comparisons analysis of ASH activation function in Appendix H (Supplementary File).
>
> ____
>
> __(Question 3) Furthermore, as one would normally use these activation functions through the network. I'm wondering if there is some effect of vanishing gradients happening, or signal reduction, due to the cancellation imposed after each layer. I could imagine that if the learned parameter would cancel 90% of the signal at each layer, this could introduce a strong loss of information.__
>
> Yes, we agree that if $k$ percentile is fixed, some effect of vanishing gradients happens. However, since rectification is performed depending on the values (contexts) of the input, and as mentioned above question, when there are many images for training, it is significantly unlikely that convolutional variables would continue to be rectified (canceled), and thus the gradient vanishing problem is not likely to be critically issued.
>
> It is commonly observed that the important features are localized when passing through the layers of a deep learning network. By observation through the experiments, a large percentile is employed in the deep depth, and thus many informative features are passed rather than rectified. Therefore, the important information and features are not filtered out due to the adaptive thresholding.
>
> ____
> __(Question 4) I'm also wondering whether this is a new approach for signal compression?.....__
>
> Yes, this study initially has started from the adaptive dropout methods. We were impressed by the reviewer’s comment and we agree with this comment. We modified the Discussion section (Supplementary File) by adding this comment as part of the future work. Thank you for the innovative comment.
>
> ____

---

### Official Review · Reviewer_djrD · 2022-07-10

**Rating:** 7
**Confidence:** 4
**Soundness:** 4 excellent
**Presentation:** 3 good
**Contribution:** 3 good

**Summary:**

In this paper, the authors present the Adaptive SwisH (ASH) activation function, which is a more generalized form of the SwisH activation function. They present the parametric and adaptive properties of the ASH function and prove the baseline threshold of the ASH function is trainable during the training phase and that the threshold value adaptively changes according to the contexts of inputs. Additionally, they prove that the ASH function does not require heavy computations compared to some other activation functions. In terms of experiments, they show that the ASH activation function outperforms other activation functions on classification, object detection, and segmentation tasks.

**Questions:**

Why were only 3 models considered for each experiment (classification, object detection, segmentation)?

**Limitations:**

The authors addressed the fact that they did not search for the best performing parameters. Instead, they justified this by stating that allowing the network to learn these parameters on its own is more similar to how human neurons work. For future work, it would be nice to see the comparison between setting the threshold for zk and letting the network learn these parameters naturally.


**Strengths And Weaknesses:**

Strengths
- proved parametric and adaptive properties of ASH function
- proved baseline threshold of the ASH function is trainable during the training phase and that the threshold value adaptively changes according to the contexts of inputs without heavy calculation
- proved ASH function is a generalized version of the SwisH activation function

Weaknesses
    Overall
    - research in the paper should be presented as a novel extension of the SwisH activation function since it is the more generalized form
    Abstract
    - should have placed the full name for the ASH activation function (Adaptive SwisH) in the "abstract" section
    Method
    - equation 10 needs more steps
    Main Result
    - Mention mathematical proofs for accuracy, sparsity, training time, and localization property as part of potential future work

---

> ### Author Response · Authors · 2022-08-02
> **Response to Reviewer djrD**
>
> ____
> __(Weakness 1) Research in the paper should be presented as a novel extension of the SwisH activation function since it is the more generalized form Abstract.__
>
> Thank you for the comments, and we modified the abstract and introduction to follow your suggestions.
>
> ____
> __(Weakness 2) Research should have placed the full name for the ASH activation function (Adaptive SwisH) in the "abstract" section.__
>
> Thank you for the comments, and we modified the abstract and introduction to follow your suggestions.
>
> ____
> __(Weakness 3) Equation 10 needs more steps.__
>
> Thank you for the comments, and we added more steps in Appendix G (Supplementary File).
>
> ____
> __(Weakness 4) Mention mathematical proofs for accuracy, sparsity, training time, and localization property as part of potential future work__
>
> We modified the related paragraph (3. Early Convergence and Optimization) and Discussion section (Supplementary File) to enhance the potential of ASH activation function.
>
> ____
> __(Question) Why were only 3 models considered for each experiment (classification, object detection, segmentation)?__
>
> We mainly applied ASH activation functions to the task in which the role of the activation function is significantly issued, even in the image generation task (previously posted on the Supplementary Files). Since vision-based analysis could exhibit the properties of activation functions in a visual manner alongside the mathematical proofs, we mainly utilized the vision-based tasks. Text- and transformer-based analysis could be further discussed for future work.
>
> ____
> __(Limitation) The authors addressed the fact that they did not search for the best performing parameters. Instead, they justified this by stating that allowing the network to learn these parameters on its own is more similar to how human neurons work. For future work, it would be nice to see the comparison between setting the threshold for z_k and letting the network learn these parameters naturally.__
>
> In the initial submission, we already explored the influence of the trainable property of ASH activation function. However, we skipped the related experiments due to the clear intuition that the fixed threshold is significantly similar to ReLU and the experiments significantly demonstrate the limitations of the fixed threshold.
>
> However, as the reviewers commented, we added the comparisons analysis of ASH activation function in Appendix H (Supplementary File). The result shows that the 50\% sampling could be fully optimized even slow. In contrast, 10\% and 90\% sampling could not be fully optimized due to the negative effects of extreme rectification by fixed threshold values.
> ____

---

### Official Review · Reviewer_oa51 · 2022-07-11

**Rating:** 7
**Confidence:** 3
**Soundness:** 3 good
**Presentation:** 2 fair
**Contribution:** 3 good

**Summary:**

The paper "Stochastic Adaptive Activation Function" successfully establishes a link between the statistics of feature selection and the recently proposed Swish-activation function.
Building upon the theoretical connection, an ASH (Adaptive Swish) non-linearity appears in the theory part.
The experimental part tests the new activation function Image net 2012, CIFAR-10, CIFAR-100, MS-COCO, and PASCAL VOC. The paper reports improved performance on every dataset and every network.


**Questions:**

- Line 17: The source code repositrory seems to be offline?
- Line 8: Does ASH mean something?
        - I found the meaning in Line 218. ASH stands for Adaptive SwisH. Perhaps this definition happens a little late in the paper?
- Line 72: What does Swish mean? Are SiLU and Swish synonymous? Could the exact definition be briefly given here?
- Equation 1: Is the ASH-function a real-valued variant of the modRelu introduced in
    [arjovski16]? The authors may be interested in adding a citation and discussing the differences.
    [arjovski16] Unitary Evolution Recurrent Neural Networks https://arxiv.org/pdf/1511.06464.pdf

- Line 172: Do we have a citation for a Z-table, for convenience?
- Table 1: What is the variance here?
- Tables 2 and 3: Is the mean accuracy shown here? What about the variance?




**Limitations:**

Limitations are not discussed at all. All numbers are perfect. Is the experimental section too good to be true?

A discussion of limitations could include:
- Potential problems that were observed during training or hyperparameter tuning. Perhaps some negative properties have been observed at some point while working with the ASH activation?
- Reasons for not reporting the variance on the mean experiment values for the Image-Net experiments?
- Perhaps some theoretical drawbacks exist that readers should be aware of?

**Strengths And Weaknesses:**

Strengths:
- The paper couples the Swish activation function to probability theory.
- The authors find improved performance for their ASH activation function across the board.
The numerical results are in line with those reported in https://arxiv.org/pdf/1710.05941;%20http://arxiv.org/abs/1710.05941.pdf for CIFAR-10.
- To the best of my knowledge, the contribution is novel. But I haven't been working looking into activation functions for some time (see confidence score below).

Weaknesses:
- The paper is hard to read at times.
- For a statistically motivated paper, it would have been neat to see variance values alongside the reported mean accuracy in the experimental section.

---

> ### Author Response · Authors · 2022-08-02
> **Response to Reviewer oa51**
>
> ____
> __(Weaknesses) For a statistically motivated paper, it would have been neat to see variance values alongside the reported mean accuracy in the experimental section.__
>
> We modified the tables related to experiments by adding variances to clarify the statistical analysis.
>
> ____
> __(Questions 1) Line 17: The source code repository seems to be offline?__
>
> Yes, but it will be online on GitHub after the publication.
>
> ____
> __(Questions 2) Line 8: Does ASH mean something? - I found the meaning in Line 218. ASH stands for Adaptive SwisH. Perhaps this definition happens a little late in the paper?__
>
> Sorry for the missing abbreviation from the abstract. We modified the manuscript.
>
> ____
> __(Questions 3) Line 72: What does Swish mean? Are SiLU and Swish synonymous? Could the exact definition be briefly given here?__
>
> Technically, SiLU and Swish are the same activation function using the sigmoid function. SiLU is firstly introduced in [1], and Swish is advanced in [2]. Mathematically, the SiLU (Sigmoid Linear Unit) and Swish exhibit similar formulas; $\text{SiLU(x)} = x\sigma(x)$ and $\text{Swish}(x) = x\sigma(\beta x)$, where $\sigma(x)$ is sigmoid function.
>
> ____
> __(Questions 4) Equation 1: Is the ASH-function a real-valued variant of the modRelu introduced in [arjovski16]? The authors may be interested in adding a citation and discussing the differences. [arjovski16] Unitary Evolution Recurrent Neural Networks https://arxiv.org/pdf/1511.06464.pdf__
>
> Thank you for introducing an interesting paper. We reviewed the paper, and we found that the modRelu is the modification of ReLU activation function but it is an odd function (symmetric). ModReLU and ASH have a similar property that they rectify inputs at specitic point, not zero like ReLU. However, the coefficient and bias of ASH activation function are trainable and depend on the context of inputs, and thus ASH exhibit more statistical significance. We added the reference to this paper, and we would like to examine quantitative analysis alongside ASH activation function in future work.
>
> ____
> __(Questions 5) Line 172: Do we have a citation for a Z-table, for convenience?__
>
> Yes, basic Statistic books say it. The reference was added to the manuscript.
>
> ____
> __(Questions 6) Table 1: What is the variance here? Tables 2 and 3: Is the mean accuracy shown here? What about the variance?__
>
> We modified the manuscript by adding 95\% Confidence Interval (C.I.) with mean values.
>
> ____
> __(Limitation 1) Potential problems that were observed during training or hyperparameter tuning. Perhaps some negative properties have been observed at some point while working with the ASH activation?__
>
> As discussed in Appendix C (Supplementary File Line 37-39), in the detection task, it is noticed that ASH activation function exhibited over-weighted localization, and thus the confidence scores were somewhat decreased. However, after the end of the training networks, the localization property increased the confidence score again, and it resulted in increasing accuracy and decreasing loss values, at the end of the training process.
>
> ____
> __(Limitation 2) Reasons for not reporting the variance on the mean experiment values for the Image-Net experiments?__
>
> We initially refer [2], so we missed the statistical reports. We modify the related tables.
>
> ____
> __(Limitation 3) Perhaps some theoretical drawbacks exist that readers should be aware of?__
>
> Perhaps the value of $z_k$. We examined many experiments using various values of $z_k$, but we could not provide a mathematical generalization. Rather, we only concluded the relation between ASH’s depth and $z_k$. $z_k$ of ASH in a deeper depth exhibited smaller values (even negative values) compared to $z_k’$ of another ASH in a shallower depth. We noticed this phenomenon by observing GRAD-CAM as shown in Supplementary Fig. 2. Additionally, we noticed that by statistically analyzing the distribution of $z_k$.
>
> Intuitively, a wide range of activations is required to recognize entire contexts of an image in a deeper depth, whereas a narrow range of activations is required to rectify features in a shallower depth in image recognition, especially a classification task. In contrast, consistent activation of the target object is required, in the segmentation or detection tasks.
>
> We appended this intuition rather than mathematical analysis in the Discussion section to highlight the characteristics of ASH activation function.
>
> ____
>
> [1] Hendrycks, Dan, and Kevin Gimpel. "Gaussian error linear units (gelus)." *arXiv preprint arXiv:1606.08415* (2016).
>
> [2] Ramachandran, Prajit, Barret Zoph, and Quoc V. Le. "Searching for activation functions." *arXiv preprint arXiv:1710.05941* (2017).

---

> > ### Comment · Reviewer_oa51 · 2022-08-09
> > **Respone to rebuttal**
> >
> > - Thank you for adding variances.
> > - An honest discussion of limitations is often a good starting point for future work. Thank you for pointing out possible limitations in the rebuttal.
> > - I have raised my rating from six to seven.
> >
> > - My only concern at this point is the source code. I don't think the results are reproducible without it. In the past, I have seen source code promises, that ultimately failed to materialize. I have adjusted my rating in good faith.

---

### Author Response · Authors · 2022-08-02
**Response to all reviewers**

We thank the reviewers' comments and we agree that they could have improved the quality of the manuscript.

In the initial submission, we skipped several experimental results that are necessary to exhibit the outstanding properties of ASH activation function. However, as the reviewers commented, we modified the missing experimental results, including statistical analysis and comparison analysis using various versions of ASH activation functions. We utilized a 95% confidence interval (95% C.I.) to exhibit statistical analysis. Finally, we modified the manuscript and the appendix files.

Again, the contributions of ASH activation function are the following:

(1) We conducted mathematical modeling for ASH in an effective form to be trainable and parametric, and thus ASH exhibits parametric and adaptive properties. The baseline threshold of ASH is trainable during the training phase, and the threshold value is adaptively changing according to the contexts of inputs without heavy calculation.

(2) We theoretically verified ASH adaptively changes its threshold alongside the stochastic distribution of inputs. This implies ASH provides outputs regarding the entire contexts of inputs, as well as leads to better feature extraction.

(3)  We theoretically verified that ASH exhibited general formula for Swish activation function and provided the mathematical explanations for the superior performance of Swish, which was empirically searched in the previous works.

(4) We experimentally showed that ASH improves the performance of deep learning models for various tasks as well as shortens training time, in terms of early convergence.

---

### Meta-Review · Area_Chair_7Bzg · 2022-08-25

**Recommendation:** Accept
**Confidence:** Certain

**Metareview:**

Reviewers appreciated the novelty of the proposed activation function, the theoretical motivation and its connection to the SwisH activation.
In terms of presentation and soundness of the results, Reviewers pointed out some weaknesses in the initial reviews for this paper. In particular, the reviews voiced some concerns with the clarity and formatting of some figures, the lack of clarity of a mathematical derivation, and most of all, issues in the presentation of the empirical results that didn't report confidence intervals allowing for an assessment of the statistical significance of accuracy differences. These weaknesses were however addressed in ways that satisfied the Reviewers in the rebuttals and subsequent versions of the paper.
Thanks to these welcome changes the paper has now garnered unanimous consensus among Reviewers that it should be accepted.

**Award:**

No

---

### Decision · Program_Chairs · 2022-09-14

Accept